# Innate extracellular vesicles from melanoma patients suppress β-catenin in tumor cells by miRNA-34a

Jung-Hyun Lee[1],*, Jochen Dindorf[1],*, Martin Eberhardt[1], Xin Lai[1], Christian Ostalecki[1], Nina Koliha[2], Stefani Gross[1], Katja Blume[1], Heiko Bruns[3], Stefan Wild[2], Gerold Schuler[1], Julio Vera[1], Andreas S Baur[1]

**Upon tumor development, new extracellular vesicles appear in circulation. Our knowledge of their relative abundance, function, and overall impact on cancer development is still preliminary. Here, we demonstrate that plasma extracellular vesicles (pEVs) of non-tumor origin are persistently increased in untreated and post-excision melanoma patients, exhibiting strong suppressive effects on the proliferation of tumor cells. Plasma vesicle numbers, miRNAs, and protein levels were elevated two- to tenfold and detected many years after tumor resection. The vesicles revealed individual and clinical stage-specific miRNA profiles as well as active ADAM10. However, whereas pEV from patients preventing tumor relapse down-regulated β-catenin and blocked tumor cell proliferation in an miR-34a–dependent manner, pEV from metastatic patients lost this ability and stimulated β-catenin–mediated transcription. Cancer-induced pEV may constitute an innate immune mechanism suppressing tumor cell activity including that of residual cancer cells present after primary surgery.**

## Introduction

Recent work suggested that most malignant cancers secrete extracellular vesicles (EVs) into the periphery that have tumorigenic properties (Skog et al, 2008; Filipazzi et al, 2012; Peinado et al, 2012; Zomer et al, 2015). Cancer EVs, such as other EVs in plasma (pEVs), contain an array of miRNAs, mRNAs, and various cellular factors and are believed to be a rich source of biomarkers (Martins et al, 2013; Properzi et al, 2013; Melo et al, 2015). Their assumed detrimental function makes them an emerging therapeutic target in cancer therapy (Vader et al, 2014).

A detailed analysis of tumor-derived EV, however, is hampered by the lack of methods that would quantify and discriminate different pEV subclasses. In addition, there is limited knowledge on the cellular origins of pEV subsets, target cell effects, and functions.

Commonly, three types of EVs are described, namely, exosomes derived from multivesicular bodies, microvesicles budding from the plasma membrane, and apoptotic bodies (EL Andaloussi et al, 2013). We described an additional type of EV, budding directly from endosomal compartments (Muratori et al, 2009; Ostalecki et al, 2016), characteristically containing numerous cytokines, chemokines, and soluble factors (hereafter termed CCF), ADAM10/17 proteases, and a surface marker set that is different from exosomes (Lee et al, 2013, 2016).

For tumor-derived EVs, which are considered to be exosomes released from multivesicular bodies, there are systemic and local effects described in vitro and in animal models (EL Andaloussi et al, 2013). This includes their ability to promote the formation of metastases by modulation of the pre-metastatic microenvironment (Peinado et al, 2012; Costa-Silva et al, 2015; Hoshino et al, 2015). In this case, tumor-derived EV have to originate from circulating or disseminated tumor cells (CTC/DTC) as metastases often arise years after the primary tumor was resected. Given the assumed limited number of residual cancer cells and rapid clearance of vesicles from circulation (Rand et al, 2006; Augustine et al, 2014), plasma concentrations of tumor-derived pEV are expected to be low in operated patients and their overall impact on cancer relapse awaits additional studies.

In plasma of melanoma patients, we previously detected ADAM10-containing pEV (Lee et al, 2013), implying that these patients harbored elevated levels of pEV similar as seen in HIV patients (Lee et al, 2016). To determine their origin and function, we systematically analyzed pEV in melanoma patients with and without tumor burden and in healthy controls. Our data revealed that these pEV were of non-tumor origin, reached high plasma concentrations, and appeared to be an innate immune response to cancer development.

## Results

### Plasma EVs are up-regulated in melanoma patients

To assess a potential increase in pEVs in melanoma patients, we first analyzed and quantified pEV-extracted miRNA levels in a

[1]Department of Dermatology, University Hospital Erlangen, Erlangen, Germany    [2]Miltenyi Biotech GmbH, Bergisch Gladbach, Germany    [3]Department of Internal Medicine V, Haematology and Oncology, University Hospital Erlangen, Erlangen, Germany

Correspondence: andreas.baur@uk-erlangen.de
*Jung-Hyun Lee and Jochen Dindorf contributed equally to this work.

similar manner as recently described (Lee et al, 2016) using the commercial miRNA chip from Agilent (see also Fig S1 and the Materials and Methods section) performed by a commercial operator (Miltenyi Biotec). To validate our centrifugation-based pEV isolation protocol, we used a marker EV containing an EBV-derived miRNA (BHRF1-2*) that was not found in human pEV miRNAs but was detectable in our miRNA microarray. After spike-in, BHRF1-2* miRNA was readily detected with comparable efficiency in four different plasma samples (Fig S1), validating our approach.

Plasma EV miRNAs were quantified by microarray from 14 melanoma patients with tumor burden, ranging from primary melanomas to skin and/or lymph-node or disseminated organ metastases (see clinical details of all patients analyzed in Tables S1 and S2). No patient suffered from other diseases or second malignancies. In comparison with age-matched healthy controls (n = 14), miRNA levels were elevated on average 6.6-fold (Fig 1A). This increase seemed related to the malignant cancer, as in two patients with an early primary melanoma (clinical stage IA), the elevated miRNA level dropped close to a level seen in controls 2 wk after surgery (Figs 1B and S2A). Conversely, no miRNA increase was observed in patients with a chronic inflammatory disease (multiple sclerosis) or semi-malignant classical (non-HIV–associated) Kaposi sarcoma (Fig 1C).

Higher miRNA levels and increased pEV numbers implied that both observations correlated. To support this observation, we measured protein concentrations in sucrose gradient fractions (1.15–1.18 mg/ml; see Fig 2F) that contained vesicles of typical shape (Fig 1D) and size (~100 ± 40 nm) (Fig S2B), with the melanoma pEV being slightly larger (mean: 129 nm) as compared with those from healthy controls (mean: 109 nm). From 30 ml plasma, we extracted 56–87 µg from patients with tumor load (metastases), but only 8–14 µg from healthy controls, revealing an average 6.9-fold increase in pEV protein concentration (Fig 1E). Particle number analysis by dynamic light scattering (ZetaView) of the same samples showed an average 10.3-fold increase in pEV numbers over controls (Fig 1F). Together, these results suggested a surprisingly strong increase in circulating pEVs in melanoma patients.

We asked whether the elevated pEV levels originated from tumor cells, as generally assumed, which would imply that these levels correlated with the tumor mass. We compared individual average pEV miRNA levels from patients with the calculated tumor mass (in cubic centimeters) based on CT scan measurements. There was no correlation between both variables, and primary melanomas with 2.8- and 0.3-mm tumor thickness (T14 and T15, Table S1) induced similar pEV levels as some patients with a tumor mass of more than 100 cm$^3$ (Fig 1G). Hence, the increased pEV levels in melanoma patients were potentially not or at least not alone secreted by the clinically assessable and/or visible tumor mass.

## Elevated pEV levels in melanoma patients after primary surgery

Although miRNA levels dropped significantly after primary surgery, they often did not fall back to levels seen in controls and remained elevated over prolonged periods (example in Fig S2C). We, therefore, compared patient's pEV levels who had R0 surgery up to 18 y before blood sampling with those bearing a tumor (hereafter termed T [tumor] patients) (see also Table S1). R0-operated patients

were subdivided into high-risk (HR) and low-risk (LR) patients, based on their clinical stage (stage II–IV versus stage I) (Boland & Gershenwald, 2016) and statistical relapse probabilities (40–95% versus 5–10% in 10 y) (Reintgen et al, 1997). In general, HR and T patients have/had a similar risk for tumor relapse. For each patient, the aggregate fold change relative to the mean of healthy controls was calculated across all miRNAs. Based on these normalized data, HR patients had much lower miRNA levels as compared with T patients, but significantly higher levels as compared with healthy controls (Fig 2A). The results showed higher statistical significance when only miRNAs were considered that were at least fourfold higher in T patients as compared with controls (Fig S2D).

In line with this result, miRNAs that were only detected in T patients, but not in controls (n = 205), were also present in HR (46/205) and LR patients (13/205) (Fig 2B). A particle measurement in gradient fractions (as in Fig 1F) revealed an average 1.9 and 5.4-fold increase in LR and HR patients over controls, respectively (Fig 2C). As HR and T patients have a similar relapse risk probability, we compared miRNA concentrations of up-regulated and randomly selected miRNAs by qPCR analysis. This revealed that both patient groups had comparable levels for these miRNAs (Figs 2D and S3), implying that the presence of a tumor mass had little influence on their presence in pEVs, at least when measured by this method. Finally, the presence of active ADAM10 was assessed in pEVs by an α-secretase activity assay (SensoLyte520) and by immunoblot, as reported previously for T patients (Lee et al, 2013). Similarly as in T patient pEVs, HR and also LR patient pEVs harbored secretase activity (Fig 2E), even 7 and 18 y after R0 surgery (Fig S4A). Confirming this result, active ADAM10 was detected by Western blot in sucrose fractions harboring pEV from T, HR, and LR patients (Fig 2F, red arrows). Conversely, age-matched controls were negative in both assays. These results suggested that melanoma-induced pEV persisted for many years after R0 surgery, although at reduced concentration levels.

We asked whether these remaining pEV levels could have originated from CTCs/DTCs, which are present when solid tumors develop (Riethmuller & Klein, 2001). We assessed the presence of melanoma-specific mutations (BRAFV600E and NRASQ61R/K) in pEV mRNAs as described previously for the EGFRvIII mutant/variant (Skog et al, 2008). The presence of mRNA with specific tumor mutations is a hallmark of tumor pEV (Melo et al, 2015). Plasma EVs were purified from 24 HR patients and controls, and mRNA was extracted and analyzed by qPCR and Sanger sequencing. The system was validated with cell culture–derived melanoma EV, using cell lines harboring either mutation (Fig S4B), and the sensitivity of the PCR assay was assessed by diluting a positive control (Fig S4C). The BRAFV600E mutation (~60% prevalence) was detected in 25% of HR patients; however, similar numbers (20%) were recorded for healthy controls. These unexpected results could be explained by BRAFV600E-positive moles (Pollock et al, 2003) shedding EVs. Conversely, the NRAS mutation (~20% prevalence) was neither detected in melanoma patients nor in controls (Fig 2G). Although these results did not exclude the presence of tumor cell (CTC/DTC)–derived pEVs, they suggested that a significant amount of pEVs present after primary tumor resection were not of tumor cell origin.

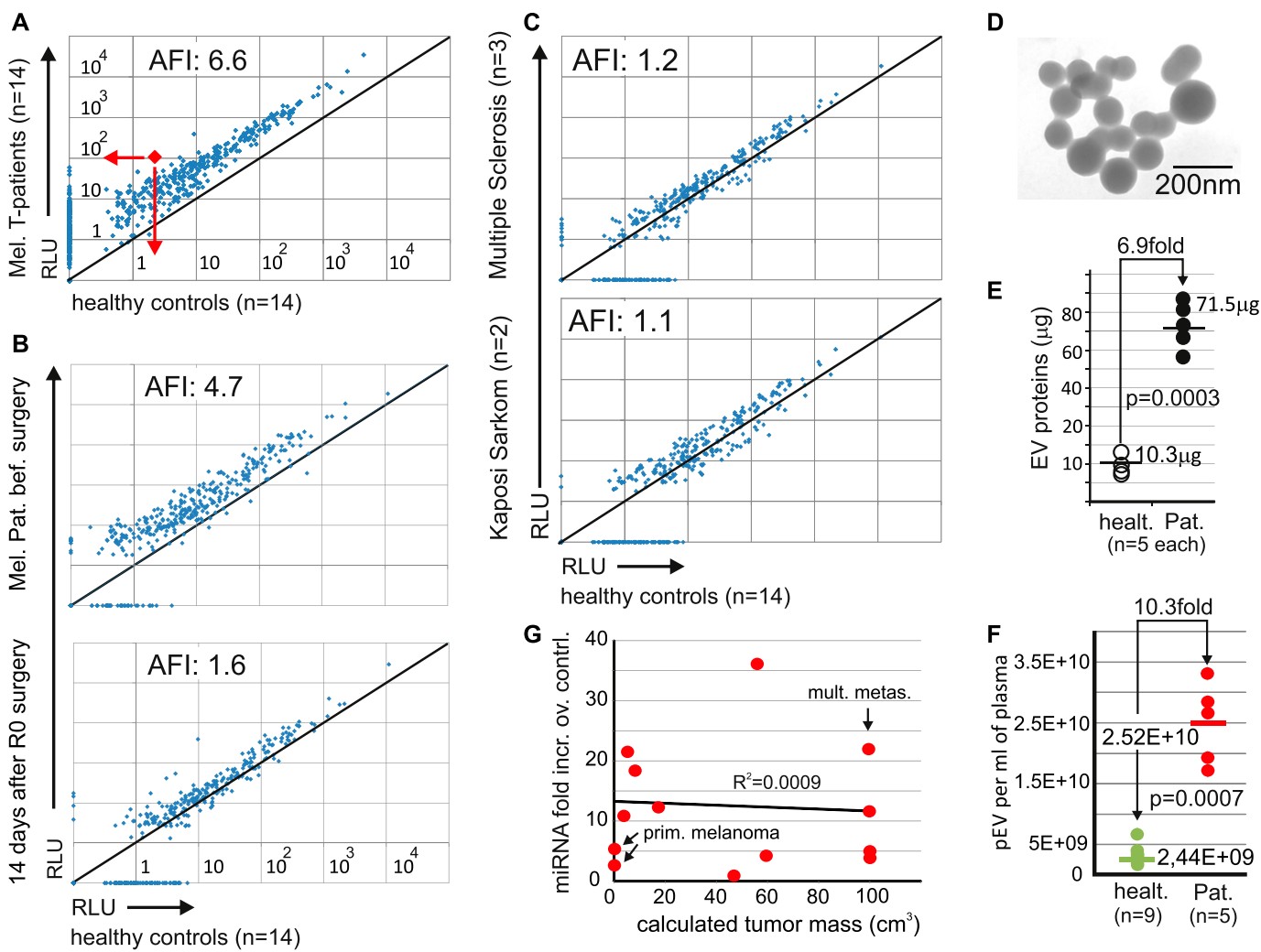

**Figure 1. Increased numbers of pEV in melanoma patients correlate with increased miRNA levels.**

**(A–C)** Analysis of average pEV miRNA levels in melanoma patients and controls. **(A)** Pairwise comparison of pEV miRNA levels/volume plasma obtained by miRNA microarrays derived from 14 melanoma patients with tumor burden (patient details in Table S1) and 14 age-matched healthy controls. Each dot represents the mean of the signal intensities of 14 patients/healthy controls for each miRNA, exemplified by the red dot and arrows. The pEV miRNAs were extracted after pEV purification by differential centrifugation from 15 ml of plasma. AFI: Average fold increase in all miRNAs over controls. **(B)** Same analysis as in (A) comparing miRNA levels in pEV from one patient with a LR melanoma (0.3 mm thickness; stage IA) before and 2 wk after R0 surgery. **(C)** Same procedure as in (A); however, pEV miRNAs were analyzed from two patients with classical (non-HIV–related) Kaposi sarcoma and three patients with multiple sclerosis and compared with the same healthy control population as in (A). **(D–F)** pEV shape, number, and size analysis. **(D)** Electromicrograph of pEVs purified from an HR melanoma patient. **(E)** Protein levels (μg) in sucrose gradient fractions (1.15–1.18 g/ml as in Fig 2F) from healthy controls (n = 5) and melanoma patients (n = 5) with tumor burden randomly selected from patients described in (A). **(F)** pEV number analysis in gradient-purified pEVs derived from melanoma patients with tumor burden (n = 5; same patients as in (E)) and nine healthy controls. The pEV numbers were assessed by ZetaView nanoparticle tracker. Based on the measurements, an average pEV concentration per ml plasma was calculated as indicated. **(G)** Correlation of tumor mass with average pEV miRNA levels. Tumor mass (in cubic centimeters) was calculated for patients with tumor burden (T1–T14, Table S1) using CT scans (performed for tumor staging) and plotted against their average pEV miRNA levels per volume plasma assessed by microarray. mult. metas., multiple metastases; Prim. melanoma, primary tumor; RLU, relative light units.

## miRNA-mediated tumor cell killing through EVs from dendritic cells but not patient's pEVs

We decided to analyze the function of melanoma-induced pEV to get a direction where these vesicles were coming from. To this end, we established a list of significantly up-regulated miRNAs that discriminated T patients from healthy controls (Fig 3A). Among those were two miRNAs (mir-215, mir-34a) belonging to a group described to activate p53 by modulating the expression of MDM2 (Pichiorri et al, 2010) and MDM4 (Mandke et al, 2012) (Fig 3B).

A clustering analysis revealed the relative up-regulation of this miRNA group in T and HR patients (Fig 3C). This suggested that melanoma-induced pEV could be tumor suppressive and eventually secreted by the immune system.

We next asked whether these miRNAs were present in EVs secreted by the immune cells. miR-215, -192, -194 (targeting MDM2), and mir-34a (targeting MDM4) were assessed by qPCR in EVs secreted by primary immune cells. Monocyte-derived mature DCs (maDCs), particularly when generated with pro-inflammatory cytokines, but much less immature DCs (imDCs), increased the

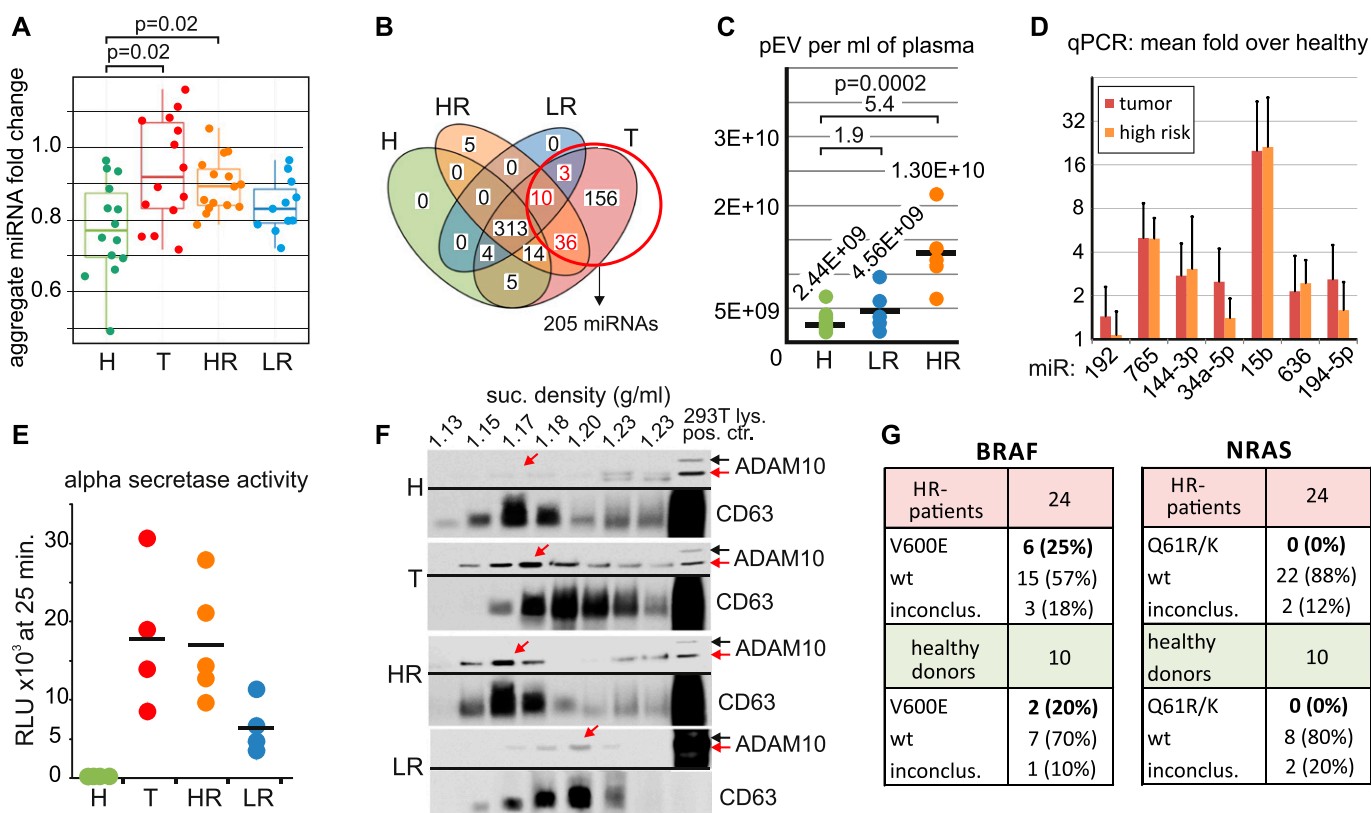

**Figure 2. Increased levels of pEVs in melanoma patients after R0 surgery.**
**(A–C)** Increased pEV miRNA levels per volume plasma in melanoma patients. **(A)** Relative levels of pEV miRNAs in healthy controls (H), melanoma patients with tumor burden (T), and in patients with LR or HR of tumor relapse after R0 surgery (Table S1). Each dot represents the average pEV miRNA level of all detected miRNAs/volume plasma from one patient and was calculated as geometric mean of the patient's miRNA fold increases relative to each miRNA's mean expression in healthy controls. **(B)** Venn diagram showing the distribution of miRNAs that discriminated healthy individuals and tumor patients (red circle) over all melanoma patient groups. **(C)** pEV number analysis in gradient-purified pEV derived from five LR and five HR patients randomly selected from patients described in Table S1 and nine healthy controls (same as in 1F). Analysis as in Fig 1F. **(D)** Quantitative PCR analysis on miRNAs up-regulated in pEVs from melanoma patients and healthy controls. Randomly selected miRNAs that were up-regulated in T patients were analyzed in parallel in pEV probes from healthy controls, HR, and T patients. Bar diagrams depict the average fold increase over healthy controls. Error bars represent the SDM of pEV samples from five representative patients or controls (Table S1). The whole procedure is detailed in Fig S3. **(E, F)** Patient's pEVs harbor ADAM10 activity. **(E)** Sucrose gradient–purified pEVs (equivalent to 1 ml of plasma) from healthy controls and melanoma patients were analyzed for alpha secretase activity using a commercial assay (SensoLyte, AnaSpec) similar as described recently (Lee et al, 2016). **(F)** Western blot demonstrating activated ADAM10 in melanoma pEVs. Sucrose gradient fractions purified from plasma (15 ml: pool from three patients [each 5 ml]) of T, HR, and LR patients containing purified pEVs were blotted for ADAM10 and the EV marker CD63. The red arrows depict activated ADAM10. **(G)** PCR amplification and Sanger sequencing of BRAF and NRAS cDNA obtained from purified HR patient pEVs. pEVs were purified from 24 HR patients and 10 healthy controls by differential centrifugation of 4 ml plasma. The table summarizes the sequencing results of the PCR amplification products for BRAF and NRAS. inconclus: inconclusive (for the presence of the mutation).

uploading of all four miRNAs into EVs and particularly miRNA-34a (Fig 3D). Conversely, the PBMC nonadherent fraction (NAF: mainly T and B cells), even when activated by PHA, did not package these miRNAs into EVs in higher amounts.

To verify a p53-targeted effect of EVs containing these miRNAs, SK-Mel32 melanoma cells were incubated with DC-derived EVs and with pEVs from melanoma patients. Only EVs from maDCs but not from imDCs or patient's pEVs induced cell death by activation of caspase-3 (Figs 3E and S5A), in a dose-dependent manner (Fig 3F). No cell killing was seen when primary cells were targeted (data not shown). To exclude nonspecific cell killing, the respective miRNAs were directly transfected into target cells, giving a similar result, whereas an unrelated miRNA (miR-17) had no effect (Fig S5B). An miRNA sponge neutralizing the miRNAs reduced cell killing by EVs from maDCs (Fig S5C), and a p53-deficient mouse lymphoma cell line (291PC p53KO) (Kovalchuk et al, 2000) was resistant to

EV-mediated cell death (Fig S5D). These results demonstrated that EVs were capable of killing tumor cells in a p53-dependent manner. Surprisingly, this was not observed for the patient's pEVs, although these pEVs contained MDM2/4 targeting miRNAs.

## Patient's pEVs modulate tumor cell proliferation

Although no target cell killing could be induced with the patient's pEVs, we observed a strong effect on tumor cell proliferation, which is another p53-mediated function (Aylon & Oren, 2007). However, whereas pEVs from T patients slightly increased cell growth relative to untreated cells, LR-/HR-derived pEVs inhibited proliferation almost completely (Fig 4A). In line with this differential effect, only the LR and HR patient pEV-treated cells showed staining with an anti-trimethyl histone H3 Lys27 antibody, implying increased histone methylation and, thus, transcriptional inhibition (Schwartz &

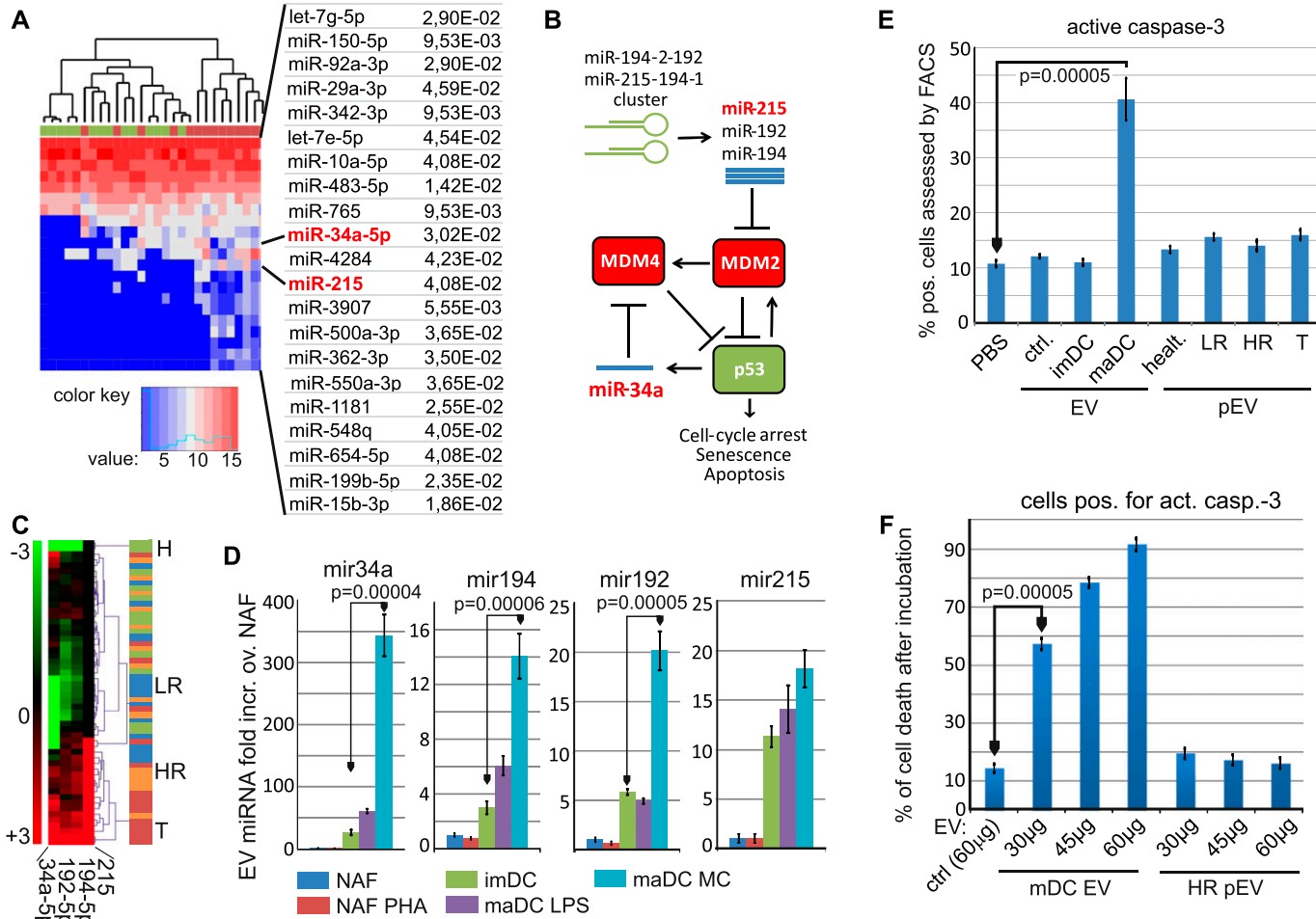

**Figure 3. DC-derived EVs, but not pEVs, kill tumor cells in a p53-dependent manner.**
**(A–C)** Presence of miRNAs regulating MDM2 and MDM4 in melanoma pEVs. **(A)** Comparison of pEV-derived miRNA levels from healthy controls (green) and T patients (red) in a heat map. The *P*-values are the adjusted results of a differential expression analysis on 14 tumor and 14 control samples (see Table S2 and Material and Methods section for details). **(B)** Cartoon depicting the regulation of p53 by miRNAs modulating MDM2 and MDM4 expressions. **(C)** Relative presence of miRNAs-34a, -192, -194, and -215 in pEVs from melanoma patients and healthy controls. **(D)** Relative presence of miRNAs-34a, -192, -194, and -215 in EVs secreted from primary immune cells. EVs were purified from culture supernatants, normalized by protein content, and assessed for the relative presence of the indicated miRNAs by qPCR (see Fig S3). Bar diagrams represent fold increase over the non-stimulated nonadherent fraction (NAF: mainly T and B cells). For each cell type, triplicate cultures were analyzed to calculate the SDM. One representative of three independent experiments is shown. maDC: DC matured by LPS or maturation cocktail. **(E, F)** Mature (matured through cytokine cocktail [Schierer et al, 2018]) DC-derived EVs, but not melanoma pEVs, kill target tumor cells in a p53-dependent fashion. **(E)** Sk-Mel32 cells were incubated with 30 µg of purified EVs, or 30 µg of pEVs, in 1 ml medium for 48 h before active caspase-3 was assessed by FACS. **(F)** Same experimental setup and read out as in (E); however, the concentration of maDC EVs and HR pEVs/ml medium was increased as indicated. For control (ctrl.), 60 µg of 293T EVs was used. Experiments in (D–F) show one representative experiment out of three, each performed in triplicates.

Pirrotta, 2007) as at least one of the underlying mechanisms for this effect (Fig 4B).

To explain the differential effect of pEVs from T- and LR/HR patients, we first confirmed the uptake of PKH26-labelled pEVs into melanoma target cells. Interestingly, T patient pEVs were incorporated more efficiently than LR or HR patient pEVs (Fig 4C); however, this did not explain their differential effect on cell proliferation.

A comparison of individual pEV miRNomes (Keller et al, 2011) by principal component analysis revealed that T patient pEV miRNomes clustered separately from LR, HR, and healthy miRNomes (Fig 4D). Indeed, those miRNAs that discriminated T patients and healthy controls differed in their concentrations in LR and HR

patient pEVs (Fig 4E). For example, whereas miRNA-34a was evenly present, miRNA-215 levels were much less increased in HR and LR patients. Thus, the overall difference of the miRNomes could at least in part explain the differential target cell effect.

### Patient's pEVs modulate the β-catenin pathway

To specify the target cell effect of patient's pEVs, we assessed 34 factors involved in cell proliferation in pEV-treated cancer cells (see antibodies in the Material and Methods section). For this approach, we used the multi-epitope-ligand-cartography (MELC) technology (Schubert et al, 2006), which allows immunostaining of one cell layer with multiple antibodies (Ostalecki et al, 2017).

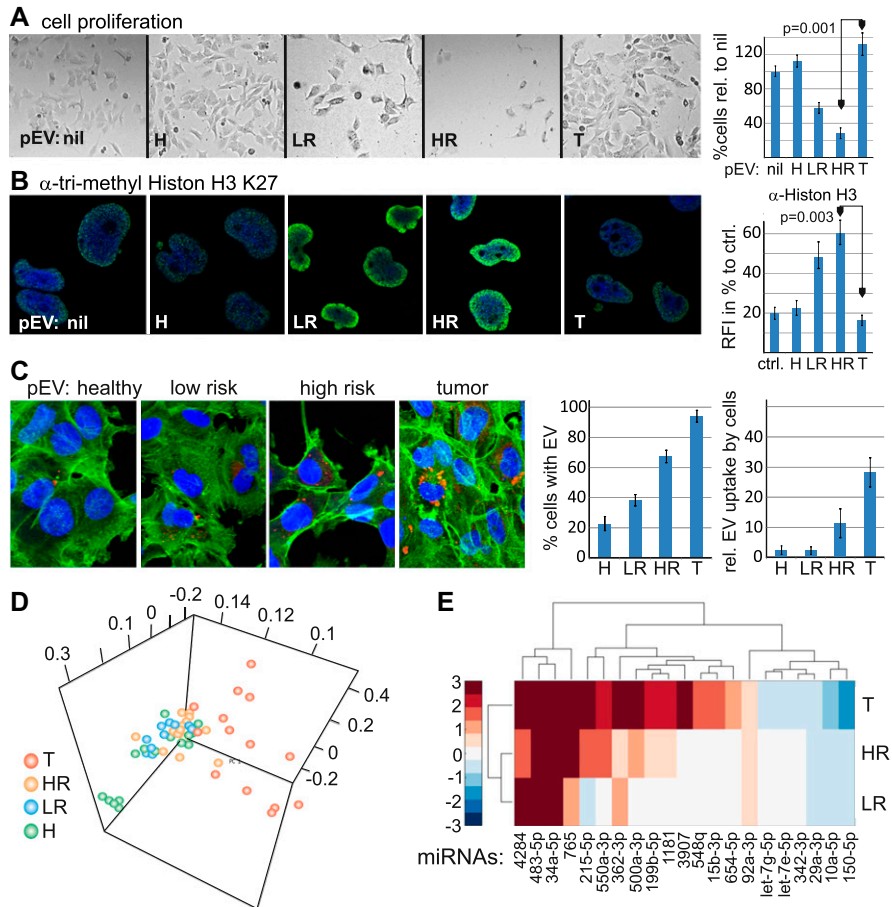

**Figure 4. Divergent function and miRNA content of LR/HR and T patient pEVs.**
**(A, B)** Melanoma pEVs modulate the proliferation of melanoma target cells. **(A)** Sub-confluent Sk-Mel32 cells were incubated with 30 µg of pEVs (in 1 ml) purified by sucrose gradient from healthy controls, LR, HR, and T patients. After 48 h, the cells were counted by trypan staining. Shown is one representative experiment of three performed using pEV from different donors. **(B)** Cells described in (A) were stained for anti-trimethyl histone H3K27. Error bars in (A) and (B) represent SDM based on triplicate cultures. **(C)** Uptake of pEV into target cells. Plasma EVs from healthy controls, LR, HR, and T patients were stained by PKH (Experimental Procedures), incubated with Sk-Mel32 target cells and analyzed for cellular uptake by counting the percentage of positive cell (out of 100) in three staining areas each. The average number of incorporated pEVs per cell was assessed by analyzing 20 positive cells in three different staining areas. Error bars represent SDM of three different areas analyzed. Experiments in (B) and (C) show one representative experiment out of three, each with a different pEV donor, and each experiment performed in triplicates. **(D, E)** T patient and LR/HR patient pEV miRNomes cluster separately. **(D)** Principal component analysis depicting the relative distance of pEV miRNA samples in LR, HR, and T patients. **(E)** Relative abundance (color coded) of pEV miRNAs that are most differently expressed in controls and T patient pEV miRNA samples (see also Fig 3A), determined for all melanoma patient groups. The color code shows log2 fold changes (red: up-regulation, blue: down-regulation). The color variation is contained in the interval (−3 to 3), meaning if the absolute log2-fold change of an miRNA is greater than 3, it shows the same color (dark blue or red).

Normalized to the expression of vimentin (Fig 5A, upper panels; see MELC analysis in the Material and Methods section), we found a significant down-regulation of β-catenin, E-Cadherin, CK2, and p21-KIP in cells treated with HR and partly LR patient pEVs, relative to control—and T patient pEV-treated cells (Fig 5A, red boxes and graphs). The latter were seemingly unaffected. All other markers differed only marginally (data not shown), including p53 (Fig 5A, lower panels).

Monocytes efficiently incorporate pEVs (Lee et al, 2013; Schierer et al, 2018), and we reasoned to see similar changes in patient's monocytes if our in vitro findings were correct. Indeed, β-catenin, both subunits of CK2a, and p21-KIP were significantly down-regulated in cells from LR and HR patients as compared with the monocytes from T patients (Fig 5B, red boxes and graphs, and Fig S6). Other markers were not affected (data not shown) (Fig S6).

For additional verification, we analyzed pEV effects on β-catenin–dependent transcription using a TCF/LEV-dependent reporter system. After transfection into SK-Mel32 cells, only T patient pEVs stimulated the reporter (Fig 5C). Preincubation (for 12 h) of transfected reporter cells with LR or HR patient pEVs partially inhibited the T pEV-mediated effect, implying that LR and HR patient pEVs were not merely inert but actively inhibited β-catenin activity (Fig 5D).

Although p53 protein levels seemed not affected by pEVs, we tested the p53-regulating miR-34a and -194 along with two other

miRNAs that were evenly (miR-92a) or differentially (miR-550a) present in patient pEVs (see Fig 4E) for their ability to modulate cell proliferation. Antagomirs to these miRNAs were transfected into melanoma target cells, which were subsequently treated with HR patient pEVs to inhibit proliferation. Only antagomirs to miR-34a abolished the HR patient pEV-induced inhibition of cell proliferation and increased histone H3 Lys27 staining (Fig 5E). Together, all results suggested that HR patient pEV inhibited tumor cell proliferation through the down-regulation of β-catenin and delivery of miR-34a.

# Discussion

Here, we report a lasting tumor-suppressive secretion of pEVs in melanoma patients that has characteristics of an innate immune response. Although we describe two effects, namely, suppression of the β-catenin pathway and inhibition of cell proliferation by miR-34a, the wide array of microRNAs in these tumor-induced pEVs point at a more complex function. Their high concentration in plasma potentially constitutes a systemic reaction that may complement immune effectors on the cellular level.

The tumor-suppressive functions were seen only with pEVs from patients without tumor load, predominantly HR patients. In HR

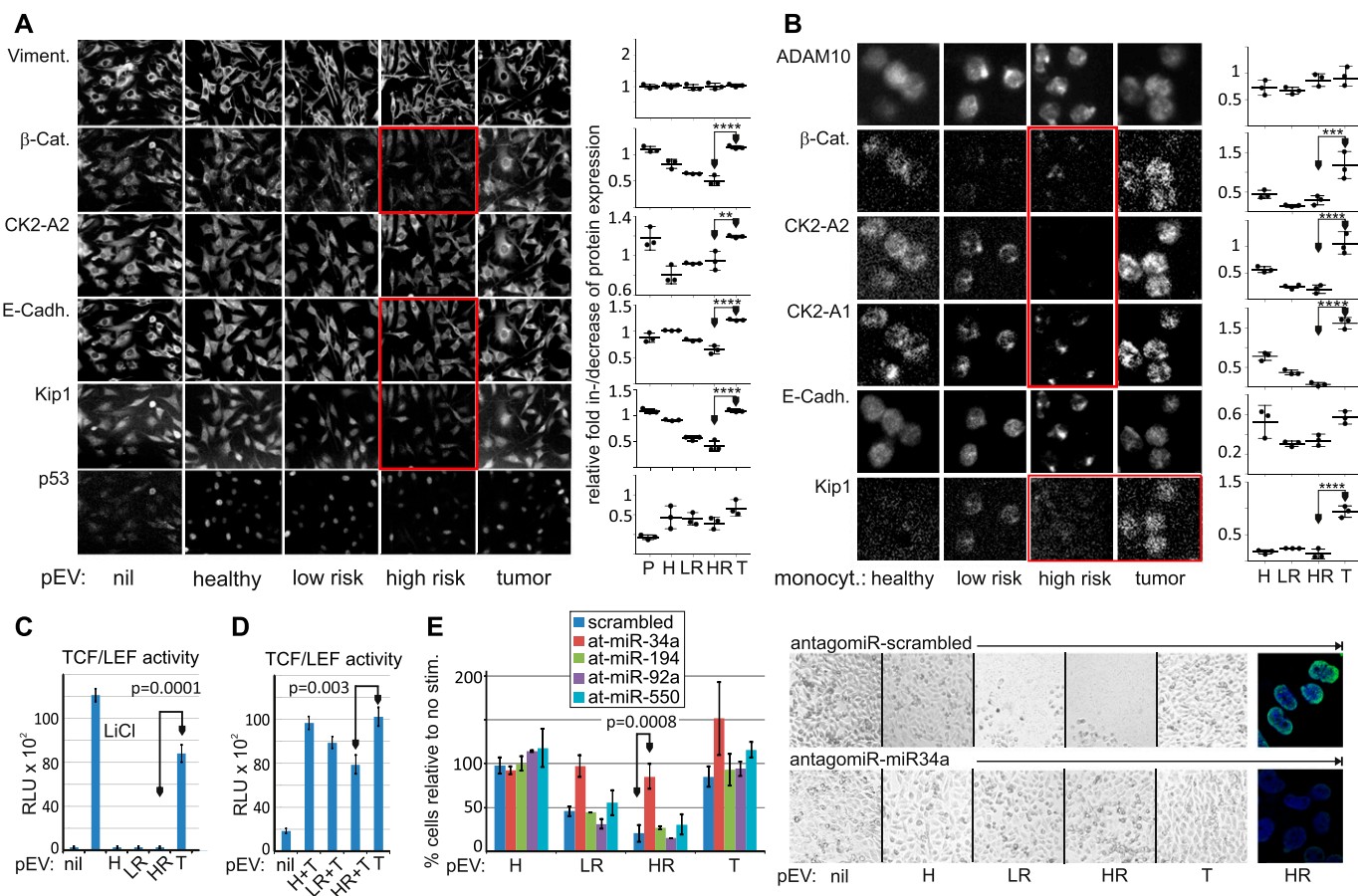

**Figure 5. Melanoma pEVs modulate the β-catenin pathway in target cells.**
**(A, B)** LR and HR, but not T patient pEVs down-regulate cell cycle and β-catenin effectors in target cells. **(A)** Sk-Mel32 cells were treated with 30 µg/ml pEVs for 48 h and subsequently analyzed by MELC technology using 34 antibodies (see MELC antibodies), including those depicted in the panel. Results with statistically confirmed differences are shown plus controls. Average protein expression levels per cell were plotted, calculating the grey scale change relative to vimentin (see also MELC analysis). **(B)** Same analysis as in (A) using monocytes from melanoma patients and controls. Average protein expression levels were calculated and plotted calculating the grey scale change relative to PI. Error bars in (A, B) represent SDM based on three visual areas analyzed. Shown in (A, B) is one of the three representative experiments, which served to calculate the statistical significance. **(C, D)** LR and HR patient pEVs inhibit, whereas T pEVs stimulate a TCF/LEF reporter. Error bars represent SDM based on analyses with pEV from three different donors. **(C)** Sk-Mel32 cells were transfected with a TCF/LEF luciferase reporter and subsequently stimulated with melanoma pEVs (30 µg/ml). Stimulation with LiCl served as positive control. Cells were analyzed after 48 h. **(D)** Same experimental setup as in (C); however, 8 h before stimulation with T pEVs, the cells were treated with H, LR, or HR patient pEVs. **(E)** HR patient pEV-mediated inhibition of melanoma cell proliferation is abolished by an antagomir to miR-34a. Sk-Mel32 cells were transfected with antagomirs against the indicated miRNAs. After 12 h, the cells were stimulated for 48 h with HR patient pEV and cell proliferation was assessed by cell counting and image documentation. In addition, two conditions were stained for tri-methyl histone H3 Lys27 as in Fig 5B. Error bars in (C–E) represent SDM based on results from three different HR pEV donors.

patients, the relative risk for tumor relapse is/was similar as for T patients, and in fact, some of the pEV characteristics were similar in both group, as, for example, the up-regulation of microRNAs not found in healthy controls (Fig 2B and D) and their content of active ADAM10 (Fig 2E). On the other hand, the number of pEVs and microRNAs in T patients was considerably higher than that in HR patients (Fig 2B). In other words, the presence of a growing tumor mass increases the number of circulating pEVs and their content. Still, this makes it difficult to identify the origin of these pEVs. Replicating tumor cells and/or CTCs/DTCs are one possible source. Immune cells directly or indirectly interacting with tumor cells are likely another source. Immune cells may sense the activity of tumor cells shed from growing tumors and/or CTCs/DTCs (Kim et al, 2009) through secreted factors, for example, cytokines and tumor

vesicles, and potentially through their secretion of RNA elements and endogenous retroviruses (Balaj et al, 2011; Kassiotis & Stoye, 2016).

Although the changing factor content in pEVs of T patients, including the miRNomes (Fig 4D), may explain a changing function, it remains unclear why inhibition of cell proliferation was conferred by LR and HR patient's pEVs but not by T patient's pEVs (Figs 4A and 5E). It is possible that the altered miRNome modulated the target effect of miR-34a; however, our own preliminary data point at an altered factor content at the protein level. One of these factors could be, for example, PD-L1 on pEVs, as shown recently (Chen et al, 2018).

Patient's pEVs inhibited tumor cell proliferation in an miR-34a–dependent manner (Fig 5E), but, unlike EV from maDC, barely

modulated p53 protein levels, although p53 was present in the tumor lines we analyzed (Fig 5A) and is generally not mutated in melanoma. miR-34a was reported to activate p53 (Hermeking, 2007). However, a correlation between miR-34a and inhibition of cancer cell proliferation was also observed in connection with the β-catenin pathway (Yan et al, 2012; Rathod et al, 2014; Chen et al, 2015). We would assume that melanoma-induced pEVs target the β-catenin pathway; however, because p53 regulation is extremely complex (Vousden & Prives, 2009), involvement of miR-34a in either pathway cannot be excluded.

The decrease in CK2a expression (Fig 5A and B) potentially complements the down-regulation of β-catenin as the kinase is a positive effector in a non-canonical β-catenin pathway (Ji et al, 2009). In addition, CK2a is an important signaling kinase phosphorylating and regulating class I HDACs and DNA methyltransferases (Liu et al, 2016). The increased histone methylation after LR and HR pEV treatment (Fig 5B) could be a consequence of its lost expression or function.

Although we have no direct evidence for a correlation between CTCs/DTCs and elevated pEV levels, our results are in accordance with the persistence rather than the elimination of CTCs/DTCs. This conclusion would support recent theories of induced cancer cell dormancy as a strategy of the immune system to control tumor relapse (Wieder et al, 2013). As we begin to understand what drives tumor cells into dormancy or senescence, one may speculate that a constant suppression of the β-catenin pathway by pEVs is at least one such mechanism. Persistent suppression could be necessary because CTCs/DTCs may emerge as proliferating cells and/or resistant to apoptosis (Meng et al, 2004; Kim et al, 2009; Aceto et al, 2014). The β-catenin pathway is of importance for the growth of melanoma tumor cells (Damsky et al, 2011) and antitumor immunity (Spranger et al, 2015). Accordingly, inhibition of β-catenin and TCF/LEF-mediated transcription is very effective in inhibiting cancer growth (Darnell, 2002).

We assume that the effects described here are very effective in vivo as cancer cells may constantly ingest circulating pEVs (Figs 4C and 6F) The complete loss of β-catenin in patient monocytes (LR/HR patients) (Fig 5B) supports this assumption. Notably, a lack of β-catenin may compromise the function of monocytes, which, for example, are relevant in controlling tumor relapse (Hanna et al, 2015). Hence, this side effect of cancer-induced pEVs could contribute to an increased risk for a second malignancy as frequently observed in cancer survivors (Caini et al, 2014; Nielsen et al, 2016).

The target cell effects of pEVs and maDC-derived EVs differed significantly for reasons that are not entirely clear, but are likely due to additional factors present at the miRNA and protein level. DCs, like other immune effector cells, kill target cells on short range after they were identified as being foreign through complex immune recognition mechanisms. In the course of this process, DCs mature and, as demonstrated here, secrete EVs with target cell killing capacity. In contrast, circulating and disseminating pEVs are possibly designed for long-range effects and are likely ingested by different cells, including monocytes (Schierer et al, 2018). In conjunction with a target cell killing capacity, this could lead to a serious autoimmune phenomenon or immune deficiency that would be counterproductive.

In summary, we provide new insights into the relevance and function of pEVs in cancer patients and suggest a potential interaction with the CTCs/DTCs. Hence, unraveling the content of pEV in cancer patients in more detail could provide crucial insight into tumor relapse. For example, a breakdown of pEV-mediated CTC/DTC suppression would allow DTCs to establish metastases in multiple locations, a phenomenon that is seen frequently upon cancer relapse.

# Materials and Methods

## Cell lines and primary cells

### Cell lines
Liver cell lines Huh7 and Sk-Hep1 (kindly provided by P. Knolle, Technische Universität München) were grown in DMEM (Sigma-Aldrich) supplemented with 10% FCS (Sigma-Aldrich) and 1% penicillin–streptomycin (Lonza). Sk-Hep1 cells were additionally maintained in 40 $\mu$M β-mercaptoethanol (Carl Roth). LX-2 cells were provided by SL. Friedman (Icahn School of Medicine, New York) and cultured in DMEM high glucose (Life Technologies) supplemented with 2% FCS, 1% penicillin–streptomycin. All cells were grown at 37°C under 5% $CO_2$.

### PBMC preparation
Leukoreduction system chambers from healthy donors were acquired after plateletpheresis. The resulting platelet-free cell sample was diluted 1:2 in PBS and the PBMC-containing buffy coat was isolated after density gradient centrifugation on Lymphoprep (Axix Shield 1114544).

### Generation of imDCs/maDCs
Monocytes were isolated from PBMCs using BD IMag Anti-Human CD14 Magnetic Particles (557769; BD Biosciences). $6.0 \times 10^6$ monocytes were seeded in a six-well plate in RPMI supplemented with 1% human serum (Sigma-Aldrich). Monocyte-derived DCs were generated adding 800 IU/ml of recombinant GM-CSF and 250 IU/ml of recombinant IL-4 (both from CellGenix). For EV isolation (see below), imDC were washed and 24 h later, the supernatant was harvested (10 ml). To generate maDCs, imDC cultures were supplemented for 24 h with LPS (100 ng/ml) or a maturation cocktail (200 IU/ml IL-1β, 1,000 IU/ml IL-6 (both from CellGenix), 10 ng/ml TNF (beromun; Boehringer Ingelheim), and 1 $\mu$g/ml Prostin E2 (PGE2; Pfizer). Subsequently, the cells were washed and EV supernatants (10 ml) were collected 24 h later for EV isolation.

### Generation of macrophages
Monocytes were separated from the nonadherent fraction (NAF) by plastic adherence on cell culture flasks and cultured in RPMI supplemented with 1% human serum and 1% of penicillin/streptomycin. On days 1, 3, 5, 7, and 9, the medium was supplemented with 800 IU/ml of GM-CSF. On day 11, the medium was removed, cells were washed, and 20 ml of RPMI supplemented with 1% of EV depleted human serum was added. After 24 h, the supernatant was harvested and EVs were isolated. For all procedures, see also Lee et al (2016).

### CTC cell line

From 30 ml blood of a melanoma patient, the CD45-positive cells were depleted using CD45 RosetteSep (Stemcell Technologies) according to the manufacturer's instructions. The remaining cells were stained with MCSP-APC and MCAM-FITC antibodies (both from Miltenyi) and DAPI (Thermo Fisher Scientific) for dead cell exclusion. MCSP-positive and/or MCAM-positive cells were then sorted on a FACS Aria SORP (BD) cell sorter and seeded in RPMI cell culture medium with 20% human pooled serum. The medium was replaced on a regular basis, and the cells showed first signs of growth after several weeks. At the time the CTC cells were obtained, the patient was tumor free.

### EV depletion of FCS and human serum for cell culture

To assure that EVs generated from cell culture were not contaminated by outside sources, heat-inactivated FCS and human serum for medium supplementation were depleted of bovine EVs by ultracentrifugation for 18 h at 110,000 $g$ and 4°C before use.

### Antibodies and reagents

Primary antibodies were used at 1–2 $\mu g \cdot ml^{-1}$ for immunoblotting, 2 $\mu g \cdot ml^{-1}$ for immunofluorescence, and 1–10 $\mu g \cdot ml^{-1}$ for MELC. The following antibodies were used for immunostaining, flow cytometry, or immunoblotting: anti-ADAM10 (mouse monoclonal, ab73402; Abcam), anti-CD63 (mouse monoclonal, 556019; BD Biosciences), anti-CD81 (mouse monoclonal, 555675; BD Biosciences), anti-haptoglobin (rabbit polyclonal, GTX 112962-25; Biozol), and anti-trimethyl histone H3 Lys27 (rabbit monoclonal, #9733; Cell Signalling). The following secondary antibodies were used: Alexa Fluor 488 goat anti-mouse and Alexa Fluor 555 goat anti-rabbit IgG (both from Life Technologies) and anti-mouse IgG-HRP conjugate and anti-rabbit IgG-HRP conjugate (both from Cell Signalling).

### Antibodies used for MELC technology

The following purified antibodies were used in this study: α-ADAM10, α-TRAF3, α-TACE, α–β-catenin, and α-Ki67 (R&D Systems); α-AGO2, α-AGO3, α-BOP1, α-CK2A2, α-CK2A1, α-DRO, α-SFRP2, α-p27$^{KIP1}$, α-TAp73, and α-TRAF1 (Helmholtz Center Munich); α-BRAF, α-PCNA, α-PPARα, α-Vimentin, and α-TACE (Santa Cruz Biotechnology); α-Bcl-2 (Dako); α-Caspase-8 and α-Cytochrome C (Biorybt); α-CD95 (Miltenyi Biotec); α-CyclinD1 and α-Notch1 (Abcam); α-E-cadherin, α-p53, α-MDM2, and α-Rac-1 (BD Pharmingen); α-Notch-2, -3, and -4 (BioLegend); α-p-Erk1/2 and α-TNF (Cell Signalling Technology); propidium iodide (Genaxxon bioscience, M3181.0010); and DAPI (4′,6-diamidino-2-phenylindole, Biomol ABD-17510).

### DNA constructs and transfections

M50 Super 8x TOPFlash (plasmid #12456; Addgene) and M51 Super 8x FOPFlash (plasmid #12457; Addgene) were gifts from Randall Moon. Plasmids were transfected with FuGENEHD Transfection Reagent (Promega) according to the manufacturer's instructions. Antagomirs were transfected using X-tremeGENE siRNA Transfection Reagent (Roche Applied Science) according to the manufacturer's

instructions. The cells were used for experiments 24–72 h after transfection.

### α-Secretase activity assay

The assay was performed essentially as described previously (Lee et al, 2016) using a commercial SensoLyte520 α-Secretase Activity Assay kit (AnaSpec 72085). Briefly, we placed sucrose gradient–purified pEVs (the equivalent of 1 ml plasma) on a 96-well, black, flat bottom plate (Greiner 655900) and added a 5-FAM (fluorophore) and QXL 520 (quencher) labelled FRET (Förster-Resonanzenergie-transfer) peptide substrate for continuous measurement of enzyme activity monitored at excitation/emission = 490 nm/520 nm by a preheated (37°C) TECAN infinite M200 Pro plate reader.

### Patient material

Plasma samples were obtained from patients attending outpatients departments at the University Hospital Erlangen after signing an informed consent. The study was approved by the local ethics committee in Erlangen (Nr. 4602). Patients were assigned to the respective study groups based on their clinical stage (Boland & Gershenwald, 2016). R0-operated patients were subdivided into HR (stage II–IV) and LR patients (stage I). T patients harbored tumor metastases (clinical stage III and IV) or primary tumors (clinical stage I–II) before surgery (see also Fig 1F and Table S1).

### Microarray analysis

The pEV were purified from platelet poor plasma and supplemented with BHRF1-2* miRNA as spike-in control (see Fig S1) and pelleted, essentially as described before (Lee et al, 2016). The pEV pellets were dissolved in 700 $\mu l$ of Qiazol, and total RNA was isolated using QIAGEN miRNeasy Mini kits (217004; QIAGEN). The extracted RNA was sent on dry ice to Miltenyi Biotec. 100 ng total RNA was concentrated to 50 ng/$\mu l$ and Cy3-labelled using Agilent's miRNA Complete Labeling and Hyb Kit (5190-0456; Agilent Technologies). After purification through Micro Bio Spin Columns (732-6221; Bio-Rad), the total RNA samples were hybridized for 20 h at 55°C to human miRNA microarrays (Agilent, Version V16, 8x60K). The microarrays were washed in Triton-containing washing buffer as recommended by the manufacturer and scanned with the Agilent's Microarray Scanner System (Agilent Technologies). The image files were analyzed and processed by Agilent Feature Extraction Software (Version 10.7.3.1).

The miRNA expression data were analyzed for logarithmic dot plots using Excel 2010 (Microsoft) and for cluster analysis with MultiExperiment Viewer Version 4.9 (MeV http://www.mybiosoftware.com/mev-4-6-2-multiple-experiment-viewer.html). http://www.tm4.org/mev.html microRNA cluster analysis was performed based on the Euclidean distance. For pairwise comparison of patient groups, means of each detected miRNA were calculated within each group and plotted on a logarithmic scale. For differential expression analysis in R, the 54-sample microarray dataset (14 healthy, 14 tumor, 15 high risk, 11 low risk) was quantile-normalized and log2-transformed. The human miRNAs on the array were then extracted, and those that failed to yield at least one intensity value above background were discarded (658 of 1,205, 54%). The limma package (Ritchie et al, 2015) was used on the remaining 547 miRNAs to fetch

out pairwise differentially expressed ones between groups. The reported logFC and adjusted *P*-values were taken from the limma results. Heat maps were generated in R with the package gplots (http://CRAN.R-project.org/package=gplots) or in MATLAB using clustergram. The Venn diagram was created in R with the package VennDiagram (https://CRAN.R-project.org/package=VennDiagram) based on the above mentioned 547 miRNAs. An miRNA was considered as occurring in a sample group if at least one of its signal intensities was above background. Average fold changes were calculated based on the quantile-normalized dataset without log-transformation. First, all intensities were normalized to the corresponding miRNA's mean intensity in the control group. All the normalized values in one sample were then averaged using the geometric mean to calculate a sample-wide fold change. The geometric mean was chosen for the second step to make the result independent of the choice of normalization group (http://doi.acm.org/10.1145/5666.5673). Principle component analysis was performed on the globally differentially expressed miRNAs (Table S2) from the log-transformed, normalized 54-sample dataset.

## Isolation, purification, and labelling of EVs and pEVs

EV and pEV purification was performed essentially as described previously (Lee et al, 2016). Briefly, cell culture supernatants were collected after 48 h and centrifuged for 20 min at 2,000 *g*, 30 min at 10,000 *g* and ultra-centrifuged for 1 h at 100,000 *g*. The pellets were resuspended in 35 ml PBS and centrifuged at 100,000 *g* for 1 h. The pellets were resuspended in 100 $\mu$l PBS and considered as EV preparations. For pEV purification, 30 ml blood plasma was diluted with 30 ml PBS and centrifuged for 30 min at 2,000 *g*, 45 min at 12,000 *g*, and ultra-centrifuged for 2 h at 110,000 *g*. The pellets were resuspended in 30 ml PBS and centrifuged at 110,000 *g* for 1 h. The pellets were again resuspended in 100 $\mu$l PBS and considered as EV preparations. For gradient purification, pEVs were diluted in 2 ml of 2.5 M sucrose, 20 mM HEPES/NaOH, pH 7.4, and a linear sucrose gradient (2–0.25 M sucrose and 20 mM HEPES/NaOH, pH 7.4) was layered on top of the EV suspension. The samples were then centrifuged at 210,000 *g* for 15 h. Gradient fractions were collected and ultra-centrifuged for 1 h at 110,000 *g*. The pellets were solubilized in SDS sample buffer or resuspended in 100 $\mu$l PBS and analyzed by immunoblotting or CCF protein array (see below). For labelling of EV with PKH (Fig 5C), we used the Sigma Mini26-1KT'' PKH26 Red Fluorescent Cell Linker Mini kit (Sigma-Aldrich) according to the manufacturer's instructions.

## Quantitative PCR amplification

The procedure is summarized in Fig S3. Reverse transcription of extracted pEV RNA was performed using the commercially available QuantiTect Reverse Transcription kit (Cat. No: 205311; QIAGEN) or TaqMan MicroRNA Reverse Transcription kit (Cat. No: 4366596; Thermo Fisher Scientific) using commercially available TaqMan MicroRNA Assays (Cat. No: 4427975; Thermo Fisher Scientific). For amplification of miRNAs, qRT-PCR was performed using TaqMan MicroRNA Assays (Cat. No: 4427975; Thermo Fisher Scientific) with a Rotor-Gene Probe PCR Kit (Cat. No: 204374; QIAGEN) according to the manufacturer's instructions on a QIAGEN Rotor-Gene Q real time PCR-cycler.

## MELC technology

The MELC technology has been described previously (Schubert et al, 2006). Briefly, a slide with cells was placed on an inverted wide-field fluorescence microscope (Leica DM IRE2; Leica Microsystems; 20× air lens; numerical aperture, 0.7) fitted with fluorescence filters for fluorescein isothiocyanate and phycoerythrin. Fluorochrome-conjugated antibodies and wash solutions were added and removed robotically under temperature control, avoiding any displacement of the sample and objective. The repetitive cyclic process of this method includes the following steps: (i) fluorescence tagging, (ii) washing, (iii) imaging, and (iv) photo bleaching; phase-contrast and fluorescence images were recorded by a high-sensitivity cooled CCD camera (Apogee KX4; Apogee Instruments; 2,048 × 2,048 pixels; 2 × binning results in images of 1,024 × 1,024 pixels; final pixel size was 900 × 900 nm). Data acquisition was fully automated.

### *MELC data analysis*
After the MELC staining procedure, the relative expression level of an antigen was determined in 10–20 representative cells by assessing the grey value intensity relative to the background. Values were obtained using the following equation:

$$RFI(AG)_X = \frac{\sum_{i=1}^{n} MGV(AG)_i}{n} - \frac{\sum_{j=1}^{m} IntDen(BG)_j}{\sum_{j=1}^{m} A_j}.$$

These values were normalized to the RFI, relative fluorescence intensity; AG, antigen; MGV, mean grey value; IntDen, integrated density; BG, background.

## Particle quantification

Sucrose-purified pEVs were diluted 1:1,000 in PBS. The pEV numbers were quantified via particle tracking analysis on a commercially available ZetaView particle tracker from ParticleMetrix using a 10-$\mu$l aliquot of the diluted samples. The concentration of pEVs was calculated based on the dilution factors.

## BRAF and NRAS PCR amplification

For detection of BRAFV600E, and NRASQ61K and Q61R mutations, pEV from HR patients were purified from 4 ml plasma by differential centrifugation as described above. Extraction of mRNA and cDNA transcription was performed essentially as outlined in Fig S3 using QIAGEN RNeasy Micro kits (Cat. No: 74004) according to the manufacturer's instructions. The amplification products were sequenced by a commercial provider (Eurofins Genomics). The following primers were used underlined in the sequence of the respective oncogene. The mutation is indicated by a capital letter, the underlined parts indicate the positions of the primers. BRAF: bp1801 5'taatatattt cttcatgaag acct**cacagt aaaaataggtgattttggtc**tagctacag**A**gaaatctcgatggagtgggtcccatcagtttgaacagt**tgtctggatccattttgtgg**at3' bp1920. NRAS: 5'bp390atagatg**gtgaaacctgtttgttggaca**tactggatacagctggac**A**agaagagtacagtgccatgagagaccaatacatgaggacaggcgaaggcttcctctgtgtatttgccatcaataa-**tagcaagtcatttgcggatatta**acctct 3'540. The system was

validated using cDNAs from primary melanoma cell line harboring either mutation (data not shown).

## Statistical analysis

Data were statistically evaluated using $t$ test or one-way ANOVA subsequently followed by Tukey's honest significant difference test when applicable.

## Data deposition

The miRNA data sets were deposited at NCBI GEO ID: GSE100508.

## Supplementary Information

## Acknowledgements

This work was supported by funds from the German Science Foundation (DFG): SFB 643 and BA961/3-1 (J-H Lee), and by the German Federal Ministry of Education and Research (BMBF) (01GU1107A) (S Wild) and MelEVIR (031L0073A) (J-H Lee). J Dindorf was supported by funds from the BMBF under grant 01GU1107A. G Schuler was supported by the IZKF Erlangen (Interdisziplinäre Zentrum für Klinische Forschung). K Blume was supported by funds from the European Union (HIVERA IRIFCURE). J Vera, M Eberhardt, and X Lai were supported by the BMBF as part of the projects eBio:miRSys (0316175A) and eBio:MelEVIR (031L0073A). J Vera is also funded by the Elan Funds of the Medical Faculty of the Friedrich-Alexander-University of Erlangen-Nürnberg (FAU) and the DFG through the project SPP 1757/1 (VE 642/1-1).

## Author Contributions

J-H Lee: data curation, formal analysis, validation, and visualization.
J Dindorf: data curation, formal analysis, validation, and methodology.
M Eberhardt: data curation and methodology.
X Lai: data curation.
C Ostalecki: data curation, visualization, and methodology.
N Koliha: data curation.
S Gross: data curation.
K Blume: data curation.
H Bruns: data curation.
S Wild: data curation.
G Schuler: funding acquisition and writing—review and editing.
J Vera: data curation and writing—review and editing.
AS Baur: conceptualization, resources, supervision, funding acquisition, validation, investigation, project administration, and writing—original draft, review, and editing.

## Conflict of Interest Statement

N.K. and S.W. are employees of Miltenyi Biotec.

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
