## [Reviewer comments · Life Science Alliance]

Life Science Alliance

Innate Extracellular Vesicles from Melanoma patients suppress b-Catenin in Tumor Cells by miRNA-34a

Andreas Baur, Lee Jung-Hyun, Jochen Dindorf, Martin Eberhardt, Xin Lai, Christian Ostalecki, Nina Koliha, Stefani Gross, Katja Blume, Heiko Bruns, Stefan Wild, Gerold Schuler, and Julio Vera

DOI: <https://doi.org/10.26508/lsa.201800205>

Corresponding author(s): Andreas Baur, University of Erlangen-Nürnberg

Review Timeline:

Submission Date:	2018-10-01
Editorial Decision:	2018-10-29
Revision Received:	2019-01-11
Editorial Decision:	2019-02-14
Revision Received:	2019-02-24
Accepted:	2019-02-25

Scientific Editor: Andrea Leibfried

Transaction Report:

October 29, 2018

Re: Life Science Alliance manuscript #LSA-2018-00205

Prof. Andreas Stephan Baur
University of Erlangen-Nürnberg
Department of Dermatology
Hartmannstrasse 14
Erlangen, Bavaria 91052
Germany

Dear Dr. Baur,

Thank you for submitting your manuscript entitled "Innate Extracellular Vesicles from Melanoma patients suppress beta-Catenin in Tumor Cells by miRNA-34a" to Life Science Alliance. The manuscript was assessed by expert reviewers, whose comments are appended to this letter.

As you will see, the reviewers think that your results are interesting. However, they also think that additional work is needed to better support your conclusions and to provide the kind of robustness needed for publication. Both reviewers provide constructive input on how to do so. We think that addressing the reviewers' individual concerns is feasible, and we would like to invite you to submit a revised version of your work for publication in Life Science Alliance, addressing the criticisms raised. Please get in touch in case you would like to discuss individual revision points further.

Thank you for this interesting contribution to Life Science Alliance. We are looking forward to receiving your revised manuscript.

Sincerely,

- A letter addressing the reviewers' comments point by point.
- An editable version of the final text (.DOC or .DOCX) is needed for copyediting (no PDFs).
- High-resolution figure, supplementary figure and video files uploaded as individual files: See our detailed guidelines for preparing your production-ready images, <http://life-science-alliance.org/authorguide>
- Summary blurb (enter in submission system): A short text summarizing in a single sentence the study (max. 200 characters including spaces). This text is used in conjunction with the titles of papers, hence should be informative and complementary to the title and running title. It should describe the context and significance of the findings for a general readership; it should be written in the present tense and refer to the work in the third person. Author names should not be mentioned.

B. MANUSCRIPT ORGANIZATION AND FORMATTING:

Full guidelines are available on our Instructions for Authors page, <http://life-science-alliance.org/authorguide>

Reviewer #1 (Comments to the Authors (Required)):

In this manuscript, Lee and Dindorf et al. demonstrate plasma extracellular vesicles (pEVs), and their miRNA content are increased in untreated and post-excision melanoma patients. The authors further show miR34a in pEVs from patients with low risk of relapse down-regulates beta-catenin and tumor cell proliferation, whereas the opposite phenotype is observed with pEVs from

metastatic patients. This manuscript provides a novel EV-mediated mechanism of tumor progression by the innate immune system. A few minor points should be addressed:

1. The authors demonstrate EVs from tumor and high-risk patients have increased miRNA levels compared to healthy patients (Fig. 2D). Are these miRNAs also increased in low-risk patients?
2. qPCR was used to detect melanoma-specific mutations in pEVs; however, the authors were unable to detect NRAS mutations and conclude the majority of pEVs are not derived from tumor cells. The authors should provide evidence the lack of detection is not due to assay sensitivity limitations. This can be accomplished by performing a dilution curve with a positive control, similar to the data provided in Fig. EV3.
3. Tumor bearing patients display increased p53-regulatory miRNAs in pEVs. The authors propose pEVs could be tumor-suppressive. In order to support the conclusion that these miRNAs have functional importance, the authors should address if these patients have deletion or loss of function mutations in p53.
4. The authors demonstrate miRNAs that regulate p53 are differentially expression in T pEVs and propose melanoma pEVs could be secreted by the immune system, but it is unclear how these miRNAs are related to immune cell secretion. This statement should be clarified in the manuscript.

Reviewer #2 (Comments to the Authors (Required)):

This manuscript contains potentially interesting findings but is suffers from repetitions and imprecise statements. The authors identify the tumor-suppressive function of circulating extracellular vesicles and provide a mechanistic explanation whereby several miRNA in the circulating exosomal milieu in tumor bearing patients target b-catenin pathway and thus suppress proliferation and promote apoptosis of the tumor cells. This is an original finding, however the quality of exosome analysis as well as the fact that the majority of the functional testing is performed in vitro diminishes the enthusiasm for this study. Below please find some of the concerns that should be addressed for the study is to be considered in good faith.

1. The experimental setup is poorly defined. Although the table EV1 is mentioned in the text, it is missing from the manuscript. Apparently, the analysis of exosomal miRNA is limited to a defined set of previously identified miRNAs. These are measured in plasma EVs of melanoma patients with significant tumor burden before and after surgery. Additionally, the patients after resection are separated into groups by high and low risk of relapse. Although it is difficult to deduce from the text, EV miRNA are significantly elevated in the groups designated as tumor and high risk prior to tumor resection and this increase is somewhat alleviated by surgery. It would be helpful, if healthy controls are always on the left and melanoma patients on the right (Figs. 1A and 1D).
2. The increased EV content is not a novel finding, and most of the panels in Fig. 1 relate to EV miRNA, so an alternative figure title should be considered. Particle size and morphology should be compared between healthy donors and patients. In general, particle characterization should be included in supplementary results.
3. The authors interpret the elevated miRNA content as the reflection of increased particle numbers; however, normalization of miRNA payload vs. particle concentration should be performed. The correlation between EV numbers/miRNA content and tumor mass does not address EV origin: it is obvious that EVs in plasma represent a mix of populations generated by multiple organs and tissues including but not limited to the tumors. Therefore melanoma and DC markers should be assessed in EV isolates by LC-MS or other means. Although feasible, general considerations used

in this section do not constitute an experimental proof. Elevated EVs could just as well be generated by residual or dormant metastatic burden. The fact that the increase in EV levels persists for 9 months argues against innate immune response, which is short-lived (Fig. EV2).

4. Although exosome levels are somewhat elevated after surgery, they appear significantly lower than in tumor-bearing patients. Since the contents and origins of EVs pre- and post-surgery could be completely different this observation does not prove the non-tumoral origin of the plasma EVs. Finally, statistical significance is not indicated for each comparison. Tukey's multiple comparisons test would be appropriate.

5. Sequencing results suggest that tumor-derived exosomes represent only a fraction of the total pEV population, which is consistent with findings by other groups and does not represent a breakthrough.

6. The elevated levels of specific miRNA, unless they are macrophage or DC-specific, do not constitute evidence in favor of the non-tumoral EV origin.

The rest of the data presented in the manuscript are of acceptable quality. Unfortunately, the connection between the characterization of patients' exosomes and the causality of tumor cell apoptosis in culture are at best tenuous. In summary, the conclusions of the manuscript should be tempered, when it comes to the origin of plasma extracellular vesicles. Vesicle characterization and analysis of their origins should be more thorough and rigorous. General considerations belong in discussion rather than in results section. The manuscript need thorough editing, to improve clarity. Unfortunately, the findings presented in this manuscript, although provocative and promising, appear too preliminary for publication

Reviewer #1 (Comments to the Authors (Required)):

In this manuscript, Lee and Dindorf et al. demonstrate plasma extracellular vesicles (pEVs), and their miRNA content are increased in untreated and post-excision melanoma patients. The authors further show miR34a in pEVs from patients with low risk of relapse down-regulates beta-catenin and tumor cell proliferation, whereas the opposite phenotype is observed with pEVs from metastatic patients. This manuscript provides a novel EV-mediated mechanism of tumor progression by the innate immune system. A few minor points should be addressed:

1. The authors demonstrate EVs from tumor and high-risk patients have increased miRNA levels compared to healthy patients (Fig. 2D). Are these miRNAs also increased in low-risk patients?

Low risk patients were not analyzed in this assay, as we wanted to compare patients in similar clinical stages with and without tumor load. In other words, we wanted to see whether high risk patients have the same increase of micro-RNAs that are typically elevated in the same risk group bearing a tumor. We made this clearer now in the text (results, page 4, second paragraph).

2. qPCR was used to detect melanoma-specific mutations in pEVs; however, the authors were unable to detect NRAS mutations and conclude the majority of pEVs are not derived from tumor cells. The authors should provide evidence the lack of detection is not due to assay sensitivity limitations. This can be accomplished by performing a dilution curve with a positive control, similar to the data provided in Fig. EV3.

We have now performed the dilution curves using a BRAF and NRAS expression plasmid. Please see supplement figure 4C and text page 4 third paragraph.

3. Tumor bearing patients display increased p53-regulatory miRNAs in pEVs. The authors propose pEVs could be tumor-suppressive. In order to support the conclusion that these miRNAs have functional importance, the authors should address if these patients have deletion or loss of function mutations in p53.

Perhaps it was a misunderstanding; we proposed that high-risk and low risk patients, but not tumor bearing patients, have a tumor suppressive effect through beta-Catenin, not through p53. Only DC-derived EV killed target tumor cells in an p53-dependent manner (Fig 3). The melanoma tumor cell lines we used for our in vitro experiments all express p53, as is evident by the staining in Fig 5A. In melanoma, p53 is usually not mutated. This is now explained in the discussion part (page 7, second paragraph)

4. The authors demonstrate miRNAs that regulate p53 are differentially expression in T pEVs and propose melanoma pEVs could be secreted by the immune system, but it is unclear how these miRNAs are related to immune cell secretion. This statement should be clarified in the manuscript.

We have now added an explanation, including citations, in the discussion part (discussion, second paragraph) explaining how and why immune cells could be secreting tumor suppressive micro-RNAs.

Reviewer #2 (Comments to the Authors (Required)):

This manuscript contains potentially interesting findings but it suffers from repetitions and imprecise statements. The authors identify the tumor-suppressive function of circulating extracellular vesicles and provide a mechanistic explanation whereby several miRNA in the circulating exosomal milieu in tumor bearing patients target b-catenin pathway and thus suppress proliferation and promote apoptosis of the tumor cells. This is an original finding, however the quality of exosome analysis as well as the fact that the majority of the functional testing is performed in vitro diminishes the enthusiasm for this study. Below please find some of the concerns that should be addressed for the study to be considered in good faith.

1. The experimental setup is poorly defined. Although the table EV1 is mentioned in the text, it is missing from the manuscript.

The referee is correct and we apologize for this mistake. Table EV 1 was supposed to contain the clinical details of every patient. It is now attached to the revised manuscript.

Apparently, the analysis of exosomal miRNA is limited to a defined set of previously identified miRNAs.

The referee is correct. Although we refer to our previous paper (Lee et al., 2016), we should have mentioned that we used a commercial micro-RNA MicroArray chip from Agilent. We added this information to the main text (results, first paragraph) and extended details in the MM section (page 10).

These are measured in plasma EVs of melanoma patients with significant tumor burden before and after surgery. Additionally, the patients after resection are separated into groups by high and low risk of relapse. Although it is difficult to deduce from the text, EV miRNA are significantly elevated in the groups designated as tumor and high risk prior to tumor resection and this increase is somewhat alleviated by surgery. It would be helpful, if healthy controls are always on the left and melanoma patients on the right (Figs. 1A and 1D).

We did change the order of the results in figure 1E and F. In graph 1A, patients were placed on the left since the increase in RLU (arrow) should face upwards.

2. The increased EV content is not a novel finding, and most of the panels in Fig. 1 relate to EV miRNA, so an alternative figure title should be considered. Particle size and morphology should be compared between healthy donors and patients. In general, particle characterization should be included in supplementary results.

We changed the title of Fig 1 to avoid a wrong impression. The additional characterization of pEV by ZetaView was added and this characterization was moved to the supplement (Figure S2B).

3. The authors interpret the elevated miRNA content as the reflection of increased particle numbers; however, normalization of miRNA payload vs. particle concentration should be performed.

We cannot retrospectively analyze micro-RNA levels and particle numbers for each patient, as there is not enough plasma left. Hence we rephrased this particular statement at the beginning of the third paragraph of the results section.

The correlation between EV numbers/miRNA content and tumor mass does not address EV origin: it is obvious that EVs in plasma represent a mix of populations generated by multiple organs and tissues including but not limited to the tumors. Therefore melanoma and DC markers should be assessed in EV isolates by LC-MS or other means.

Since there are no specific melanoma or DC markers on the protein level, such an analysis would likely not produce the desired result. Many relevant factors on the protein level are not detected by LC-MS, as for example factors below 10 kDa, like most chemokine, cytokines and soluble ligands, and these small factors are only detected if they are sufficiently charged. Furthermore, the amount of vesicles needed for such an LC-MS analysis, particularly to detect factors of low weight, is high and would exceed the amount of plasma we obtained for this study. We therefore rephrased our conclusion, leaving a potential origin of these vesicles open (page 3, end of first results section).

Although feasible, general considerations used in this section do not constitute an experimental proof. Elevated EVs could just as well be generated by residual or dormant metastatic burden. The fact that the increase in EV levels persists for 9 months argues against innate immune response, which is short-lived (Fig. EV2).

As the referee is stating correctly that residual tumor cells can persist after surgery (see also introduction) and remain for decades. Hence an innate immune response to these cells should persist as well and may correlate with the amount and activity of tumor cells derived either from the DTC/CTC cell population or a growing tumor mass. Once the tumor is removed, the population of actively replicating tumor cells is diminished, and hence the innate immune response would be expected to decrease as well. Since thin melanomas don't grow as long as thick/large melanomas they disseminate fewer DTC/CTC. Therefore the remaining innate immune response (after surgery) is expected to be less vigorous or barely measurable (Fig S2A), as compared to an immune response following the excision of a high risk melanoma. This is what we observed (Fig 2).

4. Although exosome levels are somewhat elevated after surgery, they appear significantly lower than in tumor-bearing patients.

After surgery, pEV levels are up to 5.4 fold higher in operated tumor patients than in healthy individuals, micro-RNA levels are in between 2 and 16 fold higher and ADAM 10 levels and activity is not measurable in health individuals (Fig 2). We think that this is a significant increase in pEV levels and pEV content. Supporting this conclusion these vesicles were clearly tumor suppressive as demonstrated in our functional analysis.

Since the contents and origins of EVs pre- and post-surgery could be completely different this observation does not prove the non-tumoral origin of the plasma EVs.

With the current tools and assays no one can prove that a given plasma exosome population has this or that origin, as there are no specific markers on the protein level. Even if there was, let's say a DC specific marker, there is no proof that this marker is not produced from a tumor cell or any other in vivo cell population. Hence we had to analyze this phenomenon from different angles, including a functional analysis. Tumor cells are likely not producing vesicles that suppress tumor cells. Furthermore, a self-suppressing tumor cell population (in this case it would be a CTC or DTC population) would likely not produce this vast number of excess pEV with an assumed half-life of 30 min.. It is the summary of all our findings that makes it very likely, that the

pEV levels we are describing do not originate solely from tumor cells. In fact, the referee himself/herself provides another good argument for that conclusion. He/she states that only a fraction of pEV are of tumor origin, as determined by PCR and sequencing. If this is the case, a 5-fold increase of total pEV numbers in operated tumor patients has to have a non-tumor origin.

Finally, statistical significance is not indicated for each comparison. Tukey's multiple comparisons test would be appropriate.

All statistics were performed by a professional bioinformatics group (Martin Eberhard and Julio Vera). Significance was only indicated when we found it.

5. Sequencing results suggest that tumor-derived exosomes represent only a fraction of the total pEV population, which is consistent with findings by other groups and does not represent a breakthrough.

Please see my comments above

6. The elevated levels of specific miRNA, unless they are macrophage or DC-specific, do not constitute evidence in favor of the non-tumoral EV origin.

We identified a set of four micro-RNAs in circulating pEV of operated tumor patients, but not in healthy controls that are typically down regulated in tumor cells and upregulated in activated DC, as shown in our manuscript (Fig 3). As the referee states, this is not a proof for a non-tumor cell origin (please see my comments above). However, the tumor-suppressive function of these pEV seems like a strong indication that these vesicles do not originate from tumor cells. To the best of my knowledge there are no tumor cells known, that actively suppress themselves in an autocrine fashion.

The rest of the data presented in the manuscript are of acceptable quality. Unfortunately, the connection between the characterization of patients' exosomes and the causality of tumor cell apoptosis in culture are at best tenuous.

Perhaps I misunderstood the referee, but we didn't show that pEV from patients killed tumor cells in vitro. Conversely, we demonstrated that these vesicles suppress tumor cell proliferation in vitro, which is a big difference. The latter is important as it could explain why CTC/DTC tumor cells are considered to be senescent. We do show, however, that DC-derived exosomes kill tumor cells through a set of four micro-RNAs, that are also present in patients pEV. These experiments were meant to demonstrate that EV have the principal capacity to kill tumor target cells through micro-RNAs.

In summary, the conclusions of the manuscript should be tempered, when it comes to the origin of plasma extracellular vesicles. Vesicle characterization and analysis of their origins should be more thorough and rigorous. General considerations belong in discussion rather than in results section. The manuscript need thorough editing, to improve clarity. Unfortunately, the findings presented in this manuscript, although provocative and promising, appear too preliminary for publication

REFERENCES

1. Lee JH, Schierer S, Blume K, Dindorf J, Wittki S, Xiang W, Ostalecki C, Koliha N, Wild S, Schuler G, Fackler OT, Saksela K, Harrer T, and Baur AS (2016) HIV-Nef and ADAM17-Containing Plasma Extracellular Vesicles Induce and Correlate with Immune Pathogenesis in Chronic HIV Infection. *EBioMedicine*, **6**, 103-113.

February 14, 2019

RE: Life Science Alliance Manuscript #LSA-2018-00205R

Prof. Andreas Stephan Baur
University of Erlangen-Nürnberg
Department of Dermatology
Hartmannstrasse 14
Erlangen, Bavaria 91052
Germany

Dear Dr. Baur,

Thank you for submitting your revised manuscript entitled "Innate Extracellular Vesicles from Melanoma patients suppress β -Catenin in Tumor Cells by miRNA-34a".

As you will see, reviewer #1 now supports publication, while reviewer #2 questions the extracellular vesicle miRNA data, the origin of pEVs, and the differences between high-risk and low-risk EVs. These concerns all pertain to data that were already present in the original submission, and we therefore do not expect you to address them with new experiments. However, prior to acceptance, we would like to ask you to comment on these issues in a point-by-point response and to consider text changes accordingly. Please also note that Fig3 panel H is mentioned in the legend but not present in the figure, please correct.

A. FINAL FILES:

-- Summary blurb (enter in submission system): A short text summarizing in a single sentence the study (max. 200 characters including spaces). This text is used in conjunction with the titles of

papers, hence should be informative and complementary to the title. It should describe the context and significance of the findings for a general readership; it should be written in the present tense and refer to the work in the third person. Author names should not be mentioned.

B. MANUSCRIPT ORGANIZATION AND FORMATTING:

Thank you for your attention to these final processing requirements.

Sincerely,

Reviewer #1 (Comments to the Authors (Required)):

All of my comments have been adequately addressed in the revised manuscript.

Reviewer #2 (Comments to the Authors (Required)):

This manuscript demonstrates that extracellular vesicles extracted from the plasma of melanoma patients have tumor suppressive properties and link these properties to specific miRNA found in pEVs. The manuscript has significantly improved since previous submission and offers experiments that substantiate the anti-proliferative role of miR-34a, miR-194, miR-92a and miR-550 in exosomes from patients with resected melanoma but not from tumor-bearing patients. However, the functional differences between pEVs from the low and high-risk groups has not been elucidated. Specifically, the following problems persist in this version of the manuscript:

1. The result presented in Fig. 1B (decreased pEV miRNA after tumor resection) contradicts the conclusions presented in Fig. 2A (no significant decrease after tumor resection), since significance is not indicated for these comparisons. Figures 2A and S2D show similar comparisons with opposite result. It is unclear whether the increase or decrease in exosomal miRNA post-tumor resection has any significance for the conclusions of the study as long as there are differences in miRNA content/composition of EVs.

2. The experiments in Fig. 3 (apoptosis analysis) argue against DC origin of pEV from post-resection patients, however their true origin is not ascertained or deduced. Their non-DC origin makes doubtful the innate immune role of these exosomes.

3. Since both high-risk and low-risk EVs suppress cell proliferation, downregulate b-catenin and p27 and increase 3Methyl H3K27, these results do not explain the difference in prognosis between HR and LR patients.

Therefore, the diagnostic and prognostic role of miR-34a in pEVs from melanoma patients following tumor resection remains unclear. The increase in pEVs as well as the change in composition could be due to surgery and/or wound healing process and have no significance for tumor relapse/recurrence.

Reviewer #2 (Comments to the Authors (Required)):

This manuscript demonstrates that extracellular vesicles extracted from the plasma of melanoma patients have tumor suppressive properties and link these properties to specific miRNA found in pEVs. The manuscript has significantly improved since previous submission and offers experiments that substantiate the anti-proliferative role of miR-34a, miR-194, miR-92a and miR-550 in exosomes from patients with resected melanoma but not from tumor-bearing patients. However, the functional differences between pEVs from the low and high-risk groups has not been elucidated. Specifically, the following problems persist in this version of the manuscript:

1. The result presented in Fig. 1B (decreased pEV miRNA after tumor resection) contradicts the conclusions presented in Fig. 2A (no significant decrease after tumor resection), since significance is not indicated for these comparisons. Figures 2A and S2D show similar comparisons with opposite result. It is unclear whether the increase or decrease in exosomal miRNA post-tumor resection has any significance for the conclusions of the study as long as there are differences in miRNA content/composition of EVs.

This is correct, while there were significant differences with respect to particle numbers (Figure 2C) there seemed to be no significant difference between T (tumor bearing) and HR and LR patients with respect to overall micro-RNA levels. Only when we looked at micro-RNAs at least 4fold higher, we saw a significant difference (Figure S2D) (the significance analysis was done by an independent group). However, analyzing primary melanomas before and after surgery we repeatedly noticed a visibly strong difference, particularly with thin/early melanomas, as shown in Figure 1B and S2A (these plots were established by our collaborators from Miltenyi Biotech). However, this was only an observation based on few cases. Most of the T patients in Figure 2A had metastases and were essentially relapse patients. It is possible that a T patient population with primary tumors only would reveal a significant difference in comparison to LR patients. Even though the results were not significant between T, HR and LR patients, there is a clear trend visible, which was confirmed in all the other assays and results in Figure 2. Hence we don't see a necessity to change the overall message of the paper. Furthermore, the functional effects of pEV of these patients showed significant difference between these groups (Figure 5).

2. The experiments in Fig. 3 (apoptosis analysis) argue against DC origin of pEV from post-resection patients, however their true origin is not ascertained or deduced. Their non-DC origin makes doubtful the innate immune role of these exosomes.

This is correct. Also we think that these elevated levels of pEV do not derive from mature dendritic cells. They could however derive from monocytes, macrophages or liver cells as suggested in one of our recent papers for pEV in HIV infection (Lee et al., 2018).

3. Since both high-risk and low-risk EVs suppress cell proliferation, downregulate b-catenin and p27 and increase 3Methyl H3K27, these results do not explain the difference in prognosis between HR and LR patients.

In the case of HR patients at least three factors may increase their risk for relapse, (1) the higher numbers of CTC/DTC, (2) a changing pEV factor content (e.g. PD-1) (Chen et al., 2018) and (3), the stronger suppression of beta-Catenin in monocytes (Figure 5), because the suppressive effect of pEV from HR patients was stronger as compared to the one from LR patients (Figure 4 and 5). The latter is likely an important factor as HR patients often develop a second or even third malignancy (mentioned in the discussion).

Therefore, the diagnostic and prognostic role of miR-34a in pEVs from melanoma patients following tumor resection remains unclear. The increase in pEVs as well as the change in composition could be due to surgery and/or wound healing process and have no significance for tumor relapse/recurrence

The importance of micro-RNA 34a for tumor suppression was observed by many others and there had been attempts to use micro-RNA 34a therapeutically (Mirna Therapeutics Inc.). These attempts were not successful, perhaps because the micro-RNA was not applied in vesicles. The half-life of pEV in the human body is about 30-60 min. and hence the human body makes quite an effort to maintain this secretion activity over many years. In my mind this has to have a meaning, and the interpretation described in our publication would make a lot of sense.

REFERENCES

1. Chen G, Huang AC, Zhang W, Zhang G, Wu M, Xu W, Yu Z, Yang J, Wang B, Sun H, Xia H, Man Q, Zhong W, Antelo LF, Wu B, Xiong X, Liu X, Guan L, Li T, Liu S, Yang R, Lu Y, Dong L, McGettigan S, Somasundaram R, Radhakrishnan R, Mills G, Lu Y, Kim J, Chen YH, Dong H, Zhao Y, Karakousis GC, Mitchell TC, Schuchter LM, Herlyn M, Wherry EJ, Xu X, and Guo W (2018) Exosomal PD-L1 contributes to immunosuppression and is associated with anti-PD-1 response. *Nature*, **560**, 382-386.
2. Lee JH, Ostalecki C, Zhao Z, Kesti T, Bruns H, Simon B, Harrer T, Saksela K, and Baur AS (2018) HIV Activates the Tyrosine Kinase Hck to Secrete ADAM Protease-Containing Extracellular Vesicles. *EBioMedicine*.

February 25, 2019

RE: Life Science Alliance Manuscript #LSA-2018-00205RR

Prof. Andreas Stephan Baur
University of Erlangen-Nürnberg
Department of Dermatology
Hartmannstrasse 14
Erlangen, Bavaria 91052
Germany

Dear Dr. Baur,

Thank you for submitting your Research Article entitled "Innate Extracellular Vesicles from Melanoma patients suppress b-Catenin in Tumor Cells by miRNA-34a". I appreciate your comments to the remaining reviewer concerns and it is a pleasure to let you know that your manuscript is now accepted for publication in Life Science Alliance. Congratulations on this interesting work.

DISTRIBUTION OF MATERIALS:

Again, congratulations on a very nice paper. I hope you found the review process to be constructive and are pleased with how the manuscript was handled editorially. We look forward to future exciting submissions from your lab.

Sincerely,

Andrea Leibfried, PhD
Executive Editor
Life Science Alliance
Meyerohofstr. 1
69117 Heidelberg, Germany
t +49 6221 8891 502
e a.leibfried@life-science-alliance.org
www.life-science-alliance.org